# A Survey on Learning-Based Model Predictive Control: Toward Path Tracking Control of Mobile Platforms

**Kanghua Zhang** [1], **Jixin Wang** [1,2,*], **Xueting Xin** [3], **Xiang Li** [1,*], **Chuanwen Sun** [1], **Jianfei Huang** [1]  and **Weikang Kong** [1]

1 Key Laboratory of CNC Equipment Reliability, Ministry of Education, School of Mechanical and Aerospace Engineering, Jilin University, Changchun 130022, China; zhangkh20@mails.jlu.edu.cn (K.Z.); suncw1997@163.com (C.S.); jfhuang19@mails.jlu.edu.cn (J.H.); kongwk15@mails.jlu.edu.cn (W.K.)
2 Chongqing Research Institute, Jilin University, Chongqing 401123, China
3 Research Institute, Inner Mongolia First Machinery Group Co., Ltd., Baotou 014030, China; xxtzgz@163.com
* Correspondence: jxwang@jlu.edu.cn (J.W.); xiang_li20@mails.jlu.edu.cn (X.L.)

**Abstract:** The learning-based model predictive control (LB-MPC) is an effective and critical method to solve the path tracking problem in mobile platforms under uncertain disturbances. It is well known that the machine learning (ML) methods use the historical and real-time measurement data to build data-driven prediction models. The model predictive control (MPC) provides an integrated solution for control systems with interactive variables, complex dynamics, and various constraints. The LB-MPC combines the advantages of ML and MPC. In this work, the LB-MPC technique is summarized, and the application of path tracking control in mobile platforms is discussed by considering three aspects, namely, learning and optimizing the prediction model, the controller design, and the controller output under uncertain disturbances. Furthermore, some research challenges faced by LB-MPC for path tracking control in mobile platforms are discussed.

**Keywords:** model predictive control; learning-based control; path tracking control; data-driven prediction models; uncertain disturbances; mobile platforms

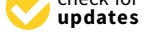

## 1. Introduction

The path tracking control is a core technique in autonomous driving. It is used to control driving in mobile platforms, such as vehicles and robots, along a given reference path, as well as to minimize the lateral and heading errors [1–3]. With the development of control theory [4,5], various advanced control algorithms [6–9] are adopted for path tracking control, including feedback linearization control [10], sliding mode control [11], optimal control [12], and intelligent control [13,14]. The design process of feedback linearization control is simple and has good response characteristics in control. Similarly, the sliding mode control has the advantages of fast response and strong robustness. The model predictive control (MPC) method in the optimal control has the ability to explicitly deal with the system constraints and extend the algorithms to multi-input multi-output systems [15]. When a reference state is introduced, the changing trend of the reference path can be added to MPC. The robust [16] and stochastic [17] model predictive controls are the main methods of dealing with uncertain systems [18]. The intelligent control achieves a better control based on self-learning, self-adaptation, and self-organization. Its performance depends on the control framework of the adopted control method [19].

Under uncertain environmental disturbances, the aforementioned methods are not fully effective in meeting the operational requirements of path tracking control. Especially in the high-speed driving environment, the complex curvature variation conditions pose a major challenge to the path tracking performance of the mobile platforms. The nonlinear motion of the mobile platforms and the complex variability of road conditions require

that the control system can intelligently achieve the established goals and ensure the real-time control of the system. The combination of machine learning (ML) and MPC has better performance in path tracking control. In [20,21], the learning-based model predictive control (LB-MPC) is applied to the path tracking control in mobile platforms. During the flight, a dual extended Kalman filter was used as a method for learning by the quadrotor to learn about its uncertainties, while an MPC was used to solve the optimization control problem. The LB-MPC technique rigorously combines statistics and learning with control engineering while providing levels of guarantees about safety, robustness, and convergence. It handles system constraints, optimizes performance with respect to a cost function, uses statistical identification tools to learn model uncertainties, and provably converges. Afterwards, the LB-MPC method was researched [22–24], and different schemes were designed [25,26]. There are various ML techniques that have been explored and applied to MPC, such as regression learning [27], reinforcement learning [28], and deep learning [29,30]. In [31], LB-MPC is reviewed from the perspective of security control, and the current applications of LB-MPC in the control field have been discussed. The LB-MPC method has been demonstrated to result in competitive high-performance control systems and has the potential to reduce modeling and engineering effort in controller design. The development process of LB-MPC is shown in Figure 1.

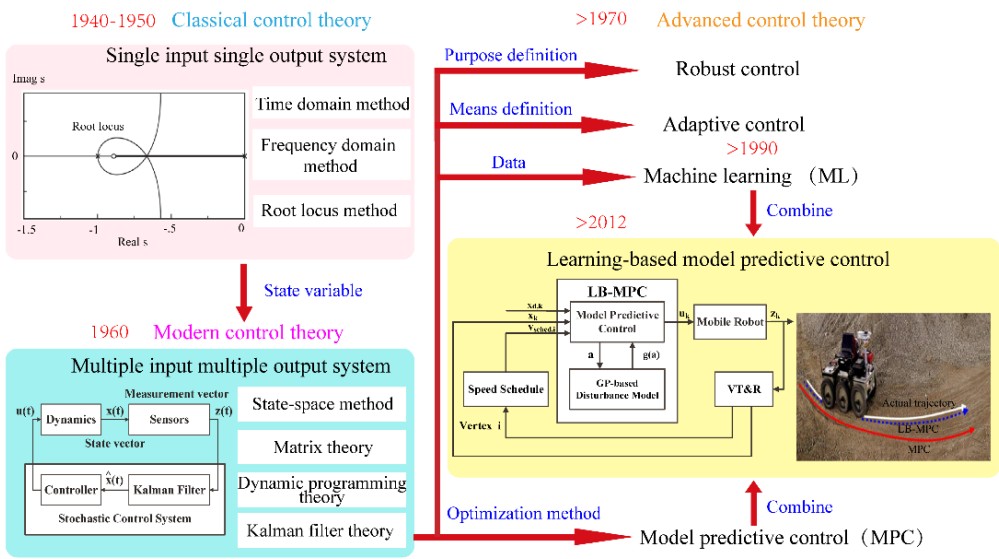

**Figure 1.** The development process of LB-MPC [4–9,32].

The paper is organized as follows. Section 2 summarizes the LB-MPC technique that combines the advantages of ML and MPC. Section 3 introduces the application of path tracking control in mobile platforms by considering three aspects. Section 4 discusses some research challenges faced by LB-MPC. Finally, Section 5 concludes this paper regarding the learning-based model predictive control technique and its application in the field of mobile platforms for path tracking control.

## 2. Overview of Learning-Based Model Predictive Control

### 2.1. Model Predictive Control

Currently, the MPC is a well-known and standard technique used for implementing constrained and multivariable control in process industries [33,34]. It provides an integrated solution for path tracking control in mobile platforms [35–37]. Unlike other optimal control methods, MPC employs a unique receding horizon technique that enables rescheduling of the optimal control strategies at each control interval to eliminate the accumulation of control errors. In mobile platforms, the MPC first establishes a prediction model. Then, all the possible states of the mobile platforms are predicted by combining the current state of the mobile platforms and the feasible control input. Finally, the state closest to the reference

state is obtained by optimizing the cost function, and the corresponding control input is attained. The MPC aims to achieve optimality in a long period of time on the basis of short-term optimization. This process involves three major steps, namely, prediction using model, rolling optimization, and feedback correction. The combination of prediction and optimization is the main difference between MPC and other control methods.

However, the MPC needs to consider the perception and response to the time-varying characteristics of a system that are caused by the uncertain disturbances. The dynamic characteristics of a system increase the uncertainty, thus affecting the performance of MPC [38,39]. In practice, model descriptions can be subject to considerable uncertainty, originating, e.g., from insufficient data, restrictive model classes, or the presence of external disturbances. Therefore, it is necessary to actively learn the uncertainties in a system and incorporate regular adaption for preserving the performance of MPC in uncertain systems.

The MPC for path tracking control in mobile platforms can be divided into linear model predictive control (LMPC), linear error model predictive control (LEMPC), nonlinear model predictive control (NMPC), and nonlinear error model predictive control (NEMPC). Figure 2 presents the comparisons of the articulated vehicle path tracking controller on the basis of LMPC, LEMPC, NMPC, and NEMPC when tracking the path with complex curvature variation [40]. The simulation results show that the performance of the NMPC controller is better as compared to LMPC, LEMPC, and NEMPC controllers. The articulated vehicle path tracking controller based on NMPC performs well in terms of stability and accuracy. However, NMPC controller needs to be further optimized in real time. Therefore, the NMPC controllers are usually used for path tracking of robots [41]. Additionally, the whole algorithm efficiency can be improved by the prediction model, cost function, and minimization algorithm.

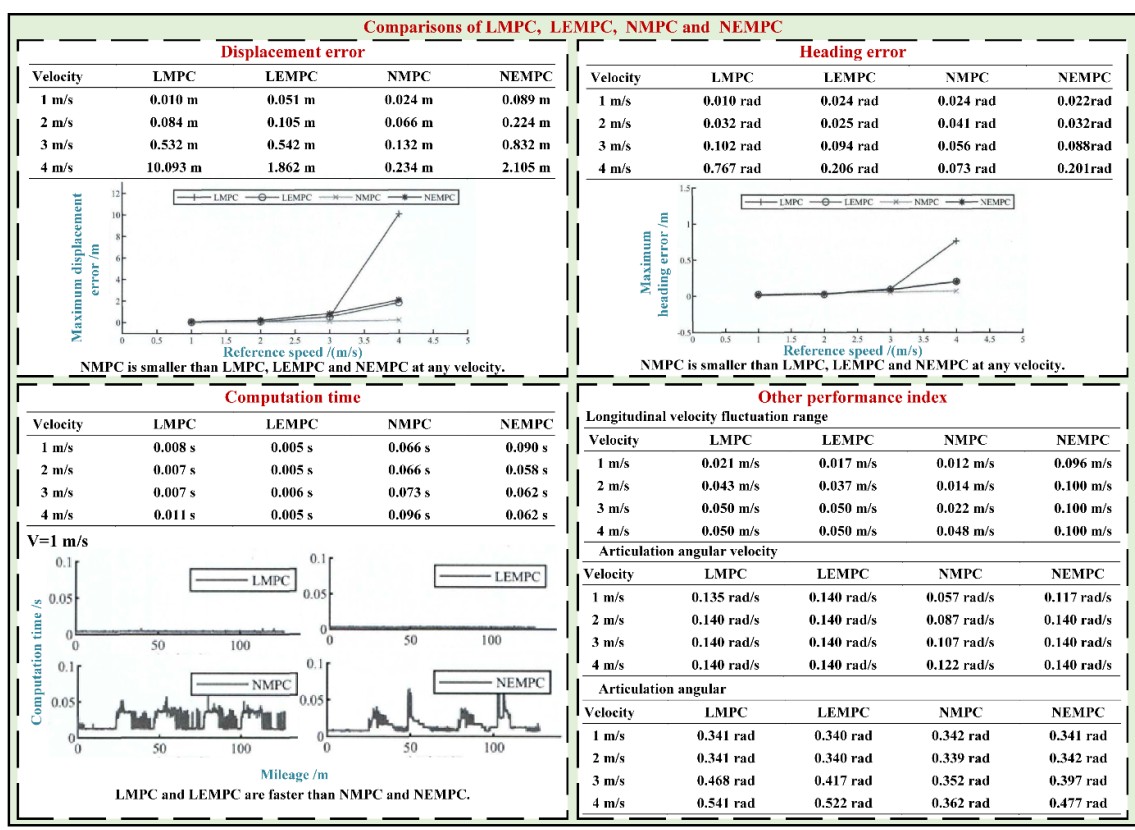

**Figure 2.** Comparisons of LMPC, LEMPC, NMPC, and NEMPC [40].

The MPC has the ability to deal with general nonlinear dynamics, hard state, input constraints, and general cost functions. Currently, solving online optimization problems has become a future research trend of MPC due to its high computational complexity.

Especially in the cases of complex physical models, numerous optimization variables, or high sampling frequency, the online optimization becomes more difficult. Moreover, it is noteworthy that the MPC is usually set up in advance by the controller and is difficult adjust in accordance with the situation. The availability of increasing computational power and sensing and communication capabilities, as well as advances in the field of machine learning, has given rise to a renewed interest in automating controller design and adaptations based on data collected during operation, e.g., for improved performance, facilitated deployment, and a reduced need for manual controller tuning.

### 2.2. Machine Learning

Recent successes in the field of ML, as well as the availability of increased sensing and computational capabilities in modern control systems, have led to a growing interest in learning and data-driven control techniques. The ML techniques have been widely used to solve complex engineering problems, such as object classification, object detection, and numerical prediction [42,43]. The ML techniques build statistical models on the basis of the training data. The resulting models are then used to predict and analyze new data on the basis of the data-driven controller [44].

Among ML techniques, neural networks have shown great success in regression problems. Additionally, neural network-based modeling outperforms other data-driven modeling methods due to its ability to implicitly detect complex nonlinearities and the availability of multiple training algorithms. In addition, ensemble learning, a machine learning paradigm that trains multiple models for the same problem, has been gaining increasing attention in recent years. By taking advantage of multiple learning algorithms or various training datasets, ensemble learning provides better predictive performance than a single machine learning model. ML techniques offer sophisticated tools for highly automated inference of prediction models from data.

The data-driven methods are effective for solving uncertain problems. During the prediction process, an uncertain model is built on the basis of the uncertain data caused by unexpected disturbances and estimation error. Then, the data-driven methods automatically use the rich information embedded in the uncertain data to make intelligent and data-driven decisions [45]. The data-driven methods are also used in the optimal design of the controller. They play an important role in learning and optimizing for the control precision and real-time response.

Iterative learning control (ILC) improves the current iterative control input signal by using the data obtained during the previous iteration [46]. As the number of iterations increases, the tracking error decreases. Therefore, ILC is widely used in ML prediction. ILC is a model-based control method that requires high model accuracy. Therefore, the learning-based prediction models are built on the basis of the data-driven methods and ILC [47]. The controller design only depends on the input and output data of the system and in not affected by the accuracy of the model. This combination improves the prediction performance of the controller.

In learning-based prediction methods, the uncertain disturbances in the mobile platforms are usually modeled as a Gaussian process (GP). The GP uses an inherent description function to estimate uncertainty [48]. The GP is a function of system state, input, and other related variables, and GP is updated on the basis of the data collected in the previous experiments. It is notable that the GP is also used to represent the dynamics of traditional systems [49]. On this basis, the GP models can be used to model complex motion modes of moving objects and compute the probability distribution of different motion modes. Afterwards, the path can be divided into different GP components to realize accurate and efficient position prediction [50]. This method does not require large number of parameters in arithmetic operations, and the position information in various motion modes can be obtained by the probability and statistical distribution characteristics of the data itself. The learning-based prediction methods adapt to various complex road conditions and improve the path tracking control performance in the mobile platforms.

As the learning-based prediction methods are more adaptable to uncertain disturbances, in the previous work, a deep learning-based approach is proposed to accurately predict the throttle and state on the basis of driving data of experienced operators [51]. The driving data comes from the field test of experimental wheel loader with sensors and GPS. Considering the time series characteristics of the process, the long short-term memory (LSTM) networks are used to extract features. The driving data are manually divided into six stages. Six LSTM networks are used for the feature extraction of six stages. The prediction of throttle and state share the same weights of LSTM in order to reduce the computational complexity. Two backward-propagation neural networks following by LSTM are used to perform regression. Two backward-propagation neural networks (BPNNs) are used to obtain the prediction results, as the throttle is controlled by the driver and the state of wheel loaders is randomly influenced by the environment. The prediction results at different stages are output by the neural networks with different parameters to improve the prediction accuracy. Wheel loaders work on different material piles during the working operation. Thus, in the previous work, two different material piles were used to study the adaptability of the prediction model. Figure 3 presents the flowchart of learning-based prediction method.

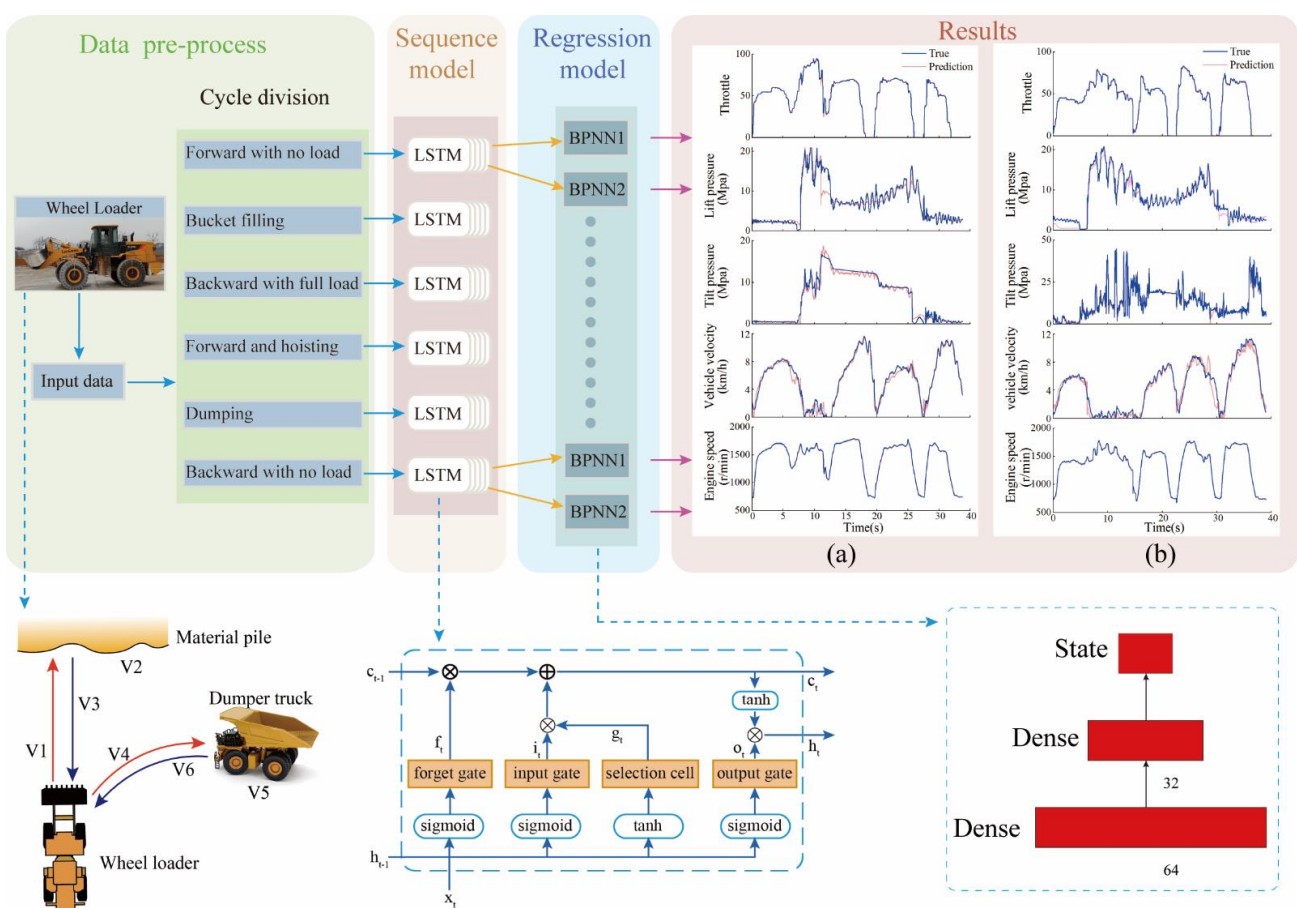

**Figure 3.** The flowchart of learning-based prediction method to accurately predict throttle and status for wheel loaders [51].

### 2.3. Learning-Based on Model Predictive Control

The MPC method provides an integrated solution for the control system with interactive variables, complex dynamics, and constraints, but it relies on the accuracy of the physical model [52]. In order for the control performance to be improved, usually more complex models are built and non-linear optimization techniques are used. The ML methods make predictions by the data-driven methods [53,54]. The ML methods build

statistical models on the basis of training data without model identification and use historical data to derive control strategy [55–57]. The ILC is applied to improve the predictive performance of the controller [58]. Theoretically, with the accumulation of valid data, the prediction ability of ML continuously improves. However, the prediction performance of ML significantly depends on the amount of training data. The perfect combination of the MPC and ML forms the LB-MPC. Applying the learning algorithm to the MPC will improve the performance of the system and guarantee safety, robustness, and convergence in the presence of states and control inputs constraints.

Most research has focused on an automatic data-based adaptation of the prediction model or uncertainty description. The feedback techniques have the ability to overcome and reduce the impact of uncertainty [59]. The LB-MPC embeds the ML method in the MPC framework to eradicate the influence of uncertain disturbances, thus improving the performance of path tracking in mobile platforms [26,60]. The LB-MPC decouples the robustness and performance requirements by employing an additional learned model and introducing it into the MPC framework along with the nominal model. The nominal model helps to ensure the closed-loop system's safety and stability, and the learned model aims to improve the tracking behaviors. The LB-MPC effectively evaluates both the current and historical effects of uncertainties, leading to superior estimating performance compared with conventional methods.

The combination of MPC and ML in nonlinear control systems has been a focus of industrial control research and development [61]. First, the LB-MPC method is used to train the model on the basis of the input data and ML, such as the GP regression method [62–65]. Second, the MPC control strategy is generated, and the calibration of MPC control parameters is performed by directly learning from the data on the basis of ML. In order for the real-time response of the controller to be realized, the sample database is trained offline by using the deep neural network [66–69]. Afterwards, non-direct measurement and the state variables for MPC are designed on the basis of ML, such as reinforcement learning [70]. Finally, the controller output is optimized on the basis of the security control framework.

In this work, we review the application of LB-MPC for path tracking control in mobile platforms, including learning and optimizing the prediction model, the controller, and the controller output in the presence of uncertain disturbances.

## 3. LB-MPC in Path Tracking of Mobile Platforms

### 3.1. Learning and Optimizing Prediction Model

The prediction model forms the basis of path tracking control. The accuracy of model determines the control performance. The simplified physical model can only be used to simulate the real mobile platform and uncertain environment to a certain extent. The prediction results are not very accurate [71]. In order for the control performance to be improved, the usual approach is to build complex physical models or use nonlinear optimization solvers. However, in the control system of mobile platforms operating in uncertain environmental disturbances, a complex physical model does not perform well in some situations. In addition, the physical model does not truly reflect the interaction between the platform and environment in real time. The prediction model is adjusted on the basis of the latest data by using data-driven ML methods [72]. For instance, the neural networks adjust their parameters on the basis of the newly acquired data [73]. The reinforcement learning realizes real-time interaction between the system and environment. The GP regression assesses the uncertainty in the residual error model to adapt the complex and dynamic working environment. The performance comparisons of the physical model, deep learning model, and reinforcement learning is presented in Table 1.

**Table 1.** The comparisons between the physical model, deep learning model, and reinforcement learning model under different evaluation parameters (ratings: none, low, medium, and high).

| Evaluation Contents | Prediction Model | | |
| --- | --- | --- | --- |
| | | Learning-Based Model | |
| | Physical Model | Deep Learning Model | Reinforcement Learning Model |
| Simulation accuracy | Low | Medium | High |
| Simulation calculation | Low | Medium | High |
| Modeling difficulty | High | Low | Medium |
| Training difficulty | None | Medium | High |
| Adaptability | Low | High | High |
| References | [71] | [29,30,66–69,74] | [28,32,70,75] |

### 3.1.1. Theoretical Basis for Modeling

The prediction models of different governing equations in MPC have a different effect on the path tracking control. The Gaussian model considers the Gaussian distribution as the model parameter [74]. It has a better control performance when dealing with periodically time-varying disturbances. The robust model predictive control (RMPC) and stochastic model predictive control (SMPC) are commonly used in uncertain systems. RMPC is especially suitable for control systems in which the stability and reliability are the primary objectives because the dynamic characteristics in the process are known and the range of uncertain factors is predictable. Moreover, RMPC does not require an accurate process model [76,77]. On the other hand, SMPC is generally intended to guarantee stability and performance of the closed-loop system in probabilistic terms by explicitly incorporating the probabilistic description of model uncertainty into an optimal control problem [78]. The disturbance rejection model predictive control (DRMPC) is based on MPC and is used for compensating the disturbances in real time. It tries to bring the disturbed system as close as possible to the calibrated system and find the nominal optimal solution by using compensation methods [79]. This is the main difference between RMPC and SMPC. In this work, both RMPC and SMPC are designed on the basis of the upper bound of disturbance. As a result, both methods are highly conservative and sacrifice some performance. The control equations of the aforementioned methods are shown in Table 2.

### 3.1.2. Data-Driven Prediction Models

The data-driven prediction models include two parts, i.e., the nominal system model and the dynamic model composed of additional uncertainties. The nominal system model is learned on the basis of the data, and it ensures the security and stability of the closed-loop system. However, due to prior uncertainty, the experimental data does not incorporate the entire state space. Consequently, the prediction accuracy of the model is not satisfactory. The uncertainty in the system arises from unmodeled nonlinearities or external disturbances and are contained in a finite set of dependent states [25]. Therefore, the uncertainty set is obtained from the data by using GP regression [80,81] or reinforcement learning [82,83], and then a dynamic model is formed with additional uncertainties. Finally, a data-driven prediction model is formed by combining the nominal system model.

The data-driven prediction models perform path tracking in mobile platforms efficiently and improve the path tracking performance in uncertain environments. An accurate vehicle model that covers the entire performance envelope of a vehicle is highly nonlinear, complex, and unrecognized. In order for uncertainties to be dealt with and for safe driving to be achieved [84], GP regression is used to obtain the residual model, which is also applied to the remote-control racing cars [48]. The trained reinforcement learning model is integrated with the controller to efficiently deal with the tracking error [85]. The inaccuracy in the prior model leads to a significant decline in the performance of MPC. The reinforcement learning based on the online model assists in learning the unknown parameters and updating the prediction model, thereby reducing the path error [86].

In the field of robotics, high precision path tracking forms the basis of robot operations. The GP of offline training is used to estimate the mismatch between the actual model and the estimated model. The extended Kalman filter estimates the mismatches in the residual model online to achieve the robot arm offset-free tracking [87]. The prediction model based on ML methods and MPC is the best solution for path tracking in cooperative autonomous unmanned aerial vehicles in the cases of different formation [88–90].

In unknown environments, such as the unstructured or off-road terrains, the robot–terrain interaction model usually does not exist. Even if such a model exists, finding the suitable model parameters is very difficult. The prior model is unable to deal with the influence of complex and dynamic terrains. The learning-based nonlinear model predictive control (LB-NMPC) algorithm includes a simple prior model and a learning disturbance model of environmental disturbance [32,75]. The disturbance is modeled as a GP function based on the system state, input, and other related variables to reduce the path tracking error of repeated traversal along the reference path. On the basis of the existing LB-NMPC algorithm, researchers have proposed a robust min–max learning-based nonlinear model predictive control [91] and a robust constraint learning-based nonlinear model predictive control (RC-LB-NMPC) method [92] to track the path of off-road terrain. The learning is used to generate low-uncertainty and non-parametric models on site. According to these models, the linear velocity and angular velocity are predicted in real time. The control framework and experimental terrain of RC-LB-NMPC are presented in Figure 4. The neural networks and model-free reinforcement learning have been used to generate walking gait for motion tasks. The neural networks are used to learn complex state transition dynamics [93]. The information learned regarding the terrain height is helpful for the robot MPC controller to track paths in uneven terrains.

**Table 2.** The governing equations for different models.

| Gaussian Model [74] | Stochastic Model [78] |
|---|---|
| Gaussian processes $r(z) \sim \mathcal{GP}(m(z), k(z, z'))$ $m(z)$: $R^{n_z} \rightarrow R$ Mean function $k(z, z\prime)$: $R^{n_z} \times R^{n_z} \rightarrow R_0^+$ Symmetric, positive semi definite covariance function $z$, $z\prime \in R^{n_z}$ Independent variables or inputs to the GP $r(z)$— At specific locations $z$— Normal distribution $R$—The set of real numbers $R^{n_z}$ —The set of n-dimensional real column vectors | A discrete-time, uncertain system $x_{k+1} = f(x_k, u_k, w_k, \theta)$ $y_k = h(x_k, v_k)$ $k \in \mathbb{N}_0$ Time index $x_k \in R^{n_x}$ —System states $u_k \in R^{n_u}$ —System inputs $y_k$ System outputs $\theta \in R^{n_\theta}$ System parameters $w_k \in R^{n_w}$ Stochastic process noise $v_k \in R^{n_v}$ Measurement noise $f(x_k, u_k, w_k, \theta)$ System state equations $h(x_k, v_k)$ Output equations |
| **Robust Optimization Model [76,77]** | **Immunity Model [79]** |
| $\widetilde{u}_f^*(t_k) = arg\ min_{u_f(\tau \mid t_k)} J_f\left(\widetilde{p}_e(t_k),\ \widetilde{u}_e(t_k)\right)$ $u_f(t) = M^{-1}\left(\theta_f(t)\right)\left[M\left(\widetilde{\theta}_f^*(t \mid t_k)\right)\widetilde{u}_f^*(t \mid t_k) + Kp_{fe}(t)\right], t \in [t_k, t_{k+1})$ $\widetilde{u}_f^*(t_k)$ Optimal predictive control sequence $u_f(t)$ Actual control signal $J_f\left(\widetilde{p}_e(t_k),\ \widetilde{u}_e(t_k)\right)$—The cost functions in model predictive control $\widetilde{p}_e(t_k)$—The tracking error of the system $\widetilde{u}_e(t_k)$—The control of input error $\theta_f(t)$—The angle of system state $p_{fe}(t)$ State of the actual system | $\widetilde{u}_f^*(t_k) = arg\ min_{\widetilde{u}_f(\tau \mid t_k)} J\left(\widetilde{p}_e(t_k),\ \widetilde{u}_e(t_k)\right)$ $u_f(\tau) = \kappa\left(\widetilde{u}_f^*(t_k \mid t_k), \widehat{d}_v(t_k)\right)$ $\widetilde{u}_f^*(t_k)$—Optimal control sequence $u_f(\tau)$—Composite control signals $\widehat{d}_v(t_k)$—Estimation of actual disturbance |

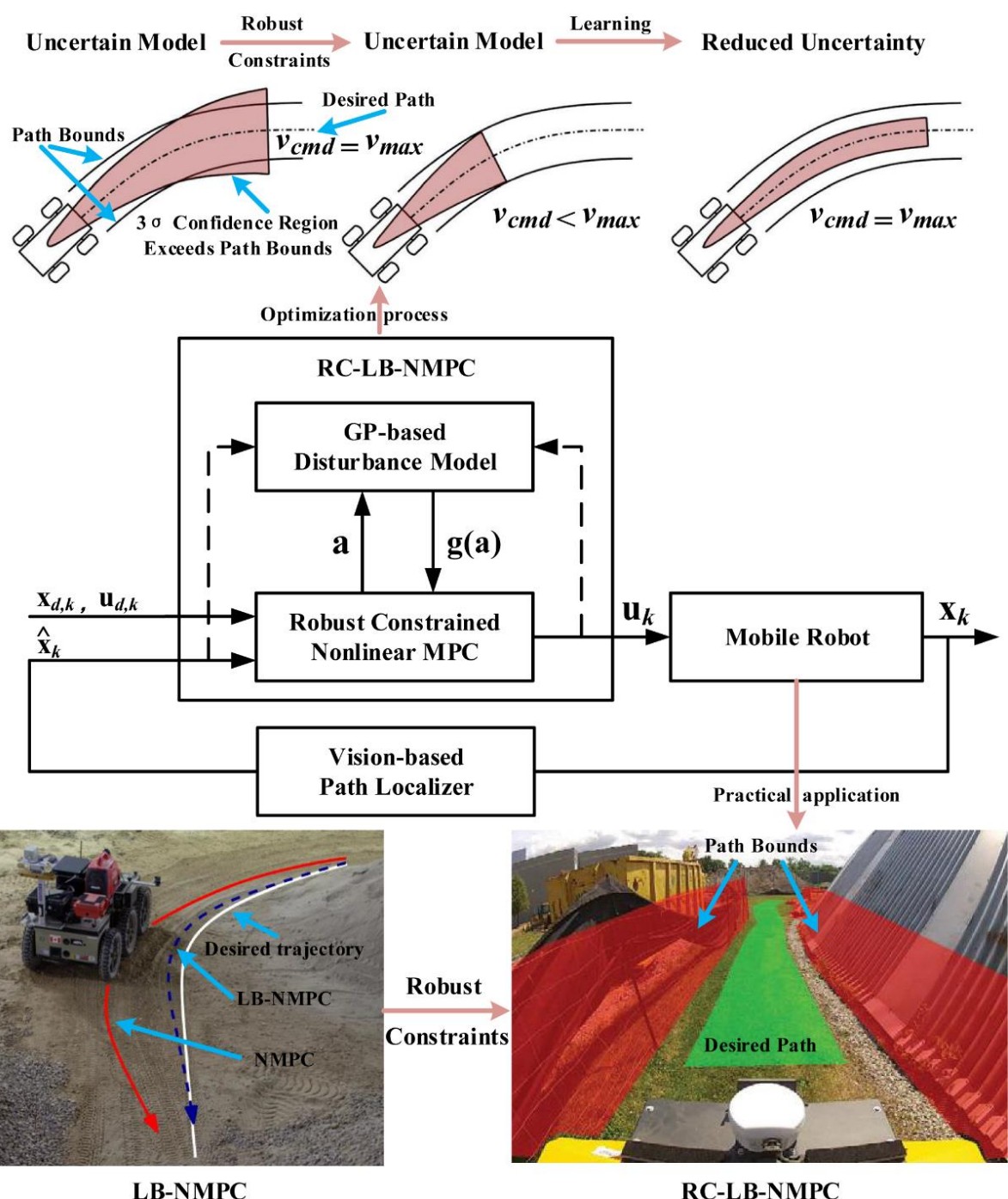

**Figure 4.** The control framework and experimental terrain of RC-LB-NMPC method [32,75,92].

### 3.1.3. Prediction Model Pre-Training Based on Transfer Learning

In ML, a large amount of data is often required to learn complex features. The process of acquiring the data for training the mobile platform prediction model is a laborious task. It is noteworthy that as soon as the feature space or feature distribution of the test data changes, the performance of model degrades. As a result, new data must be collected to retrain the model for enhancing the performance.

The transfer learning methods exploit the knowledge accumulated from data in auxiliary domains to facilitate the predictive modeling consisting of different data patterns in the current domain [94]. These methods include sample transfer, feature transfer, model transfer, and relation transfer. In natural language processing and computer vision, the

pre-trained models created for a specific task are used to solve problems of a different domain [95]. However, there are not many neural networks available for path tracking control in mobile platforms. Therefore, the simulation dataset for model pre-training is obtained by building simulation models, such as dynamics. The data obtained from the real mobile platform is used to fine-tune the prediction model. As a result, the requirement for real data is reduced. The degree of similarity between the simulation and real data significantly influences the results of pre-training.

In multibody dynamics simulation (MBS) analysis, there are three challenges, namely, high modeling difficulty, high computational complexity, and restricted solver. The MBS based on deep learning network (MBSNet) is applied to vehicle tracking systems [96]. This model is robust in the presence of different track irregularities. The MBSNet accurately and quickly predicts the low-frequency components of dynamic response. The technical flowchart of MBSNet is presented in Figure 5.

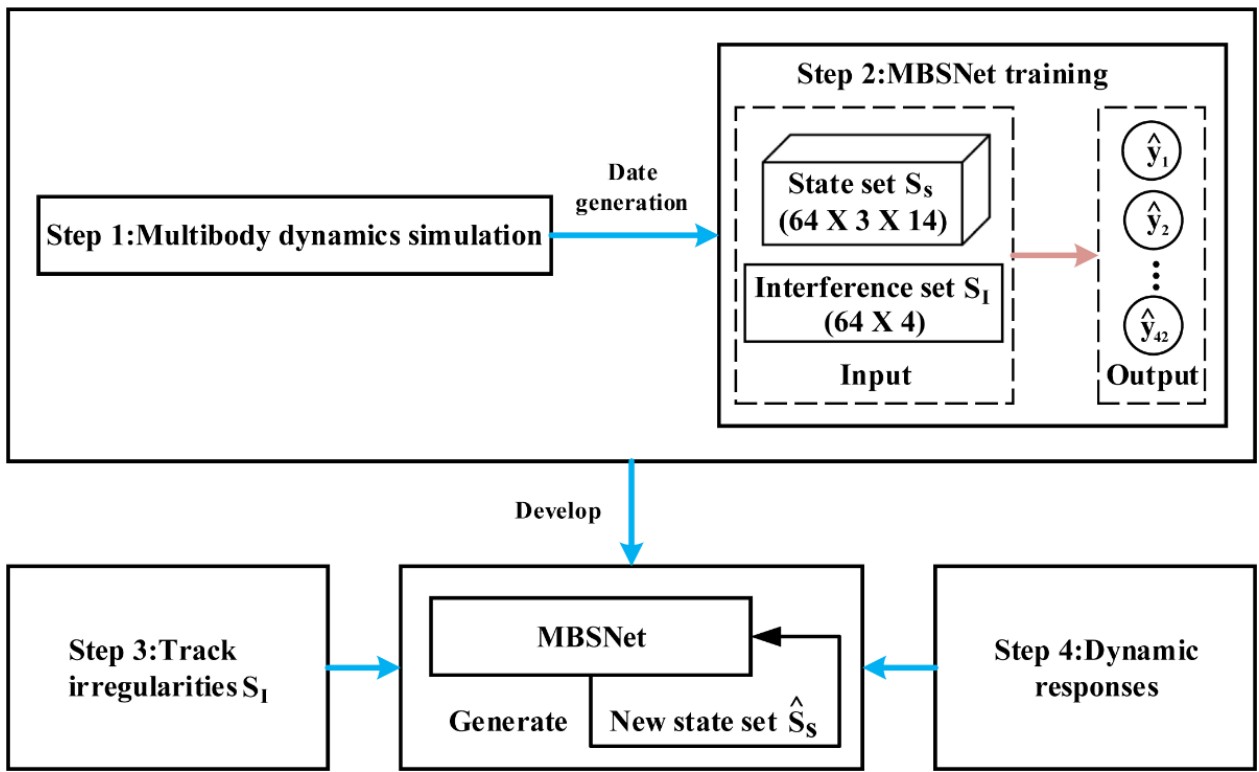

**Figure 5.** The technical flowchart of multi-body dynamics simulation based on deep learning network [96].

The end-to-end training methods use the original sensor data as the input and train the output directly by commands. These methods decrease the volume of training data because they are only trained on real devices or simulations without experiments. The convolutional neural network (CNN) is used as an image embedder for estimating the current and future position of the vehicle relative to the path [97]. As presented in Figure 6, a region randomization method is proposed to generate different types of paths with random curvature and length, initial lateral, and heading errors for the vehicle during simulation. This method covers all the possible training scenarios. In addition, this method prevents the network from overfitting, which makes it more universal as compared to other similar methods. The mobile platforms follow the path smoothly and slow down where the curvature is larger.

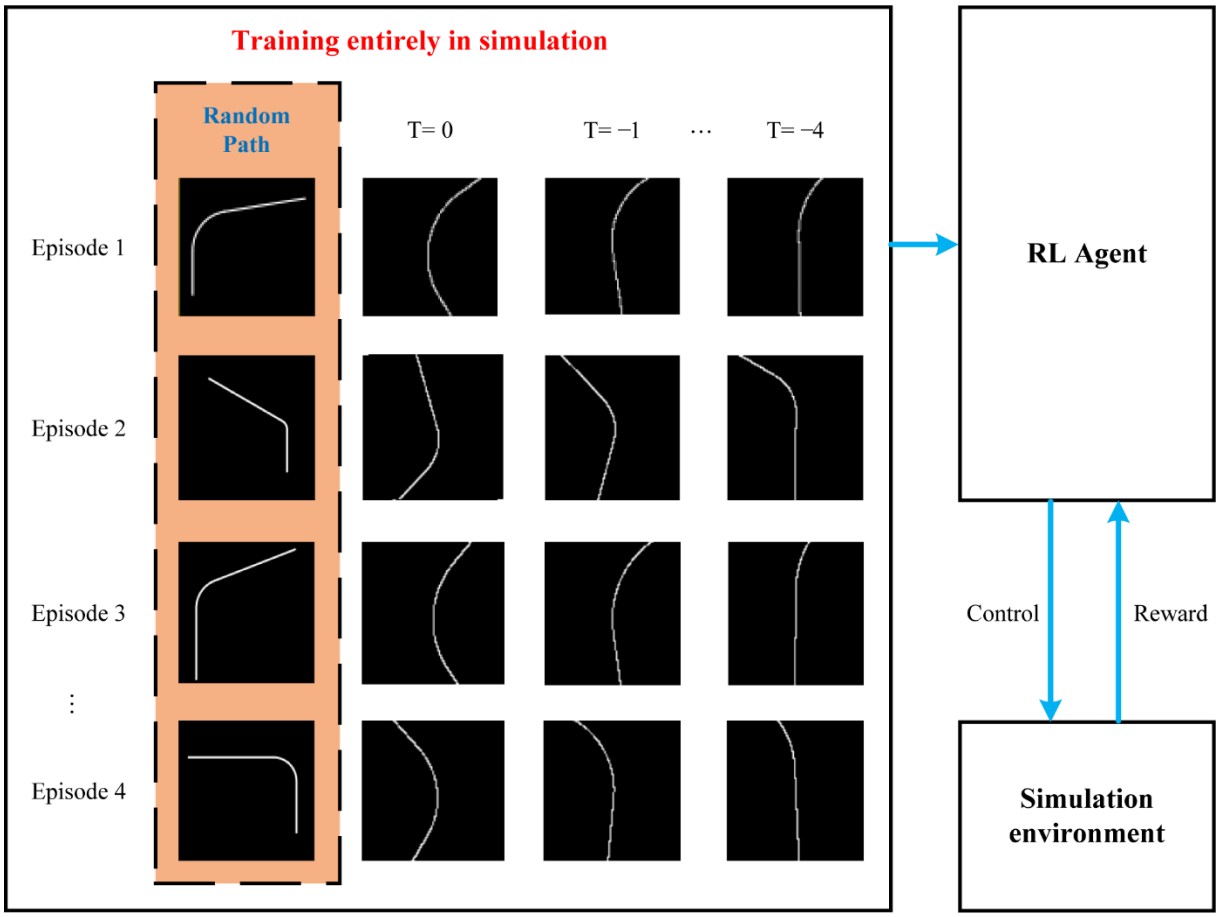

**Figure 6.** The path randomization for training [97].

In cases of uncertain disturbances, the data-driven prediction models have a better environmental adaptability as compared to the traditional physical models. The transfer learning is used to pre-train the prediction model in the simulation environment, which solves the problem of high demand for real training data. The transfer learning also improves the reliability of the data-driven prediction models, which is the basis of LB-MPC.

### 3.2. Learning and Optimizing for Controller

It is important to control the mobile platforms to follow a complex curvature path. The prediction model is the core element of the model predictive controller, and other elements, such as cost functions, constraints, and real-time response, also have a significant impact on the final closed-loop control performance.

#### 3.2.1. Learning and Optimizing for Control Precision

The model predictive controller optimizes the cost function under various constraints. The optimization design of the controller precision is based on the parameterization of the cost function and constraints. The cost function and the constraint condition can be parameterized into the function of the system state variables, application input, and random variables. Table 3 shows a parameterized version of an MPC problem, including a parameterization of the cost function $l(x_i, u_i, \theta_t)$ and constraints $\mathcal{X}(\theta_{\mathcal{X}}), \mathcal{U}(\theta_{\mathcal{U}})$ [31]. This is convenient for subsequent learning and optimization of the controller parameters under uncertain disturbances.

**Table 3.** The parameterized version of an MPC problem [31].

| Evaluation Contents | Expression Formula |
|---|---|
| The system dynamics in discrete time | $x(k+1) = f_t(x(k), u(k), k, w(k), \theta_t)$<br>$x(k)$ The system state<br>$u(k)$ The applied input at time $k$<br>$w(k)$ The noise in the system<br>$\theta_t$ The parametric uncertainty of the system |
| The cost or objective function | $J_t = E\left(\sum\limits_{k=0}^{\overline{N}} l_t(x(k), u(k), k)\right)$<br><br>$\overline{N}$ The possibly infinite horizon |
| Both the system dynamics and the prediction model in LB-MPC under uncertain disturbances | $f(x, u, k, \theta, w) = f_n(x, u, k) + f_1(x, u, k, \theta, w)$<br>$f_n$ The nominal system model<br>$f_1$ The additive learned term |
| The parameterization of the cost function and constraints in optimal control problem | $U^* = \underset{U}{argmin} \sum\limits_{i=0}^{N} l(x_i, u_i, \theta_l)$<br><br>subject to $x_{i+1} = f\left(x_i, u_i, \theta_f\right)$<br>$U = [u_0, \dots, u_N] \in \mathcal{U}(\theta_\mathcal{U})$<br>$X = [x_0, \dots, x_N] \in \mathcal{X}(\theta_\mathcal{X})$<br>$x_0 = x(k).$<br>$l(x_i, u_i, \theta_l)$ The cost function<br>$\mathcal{X}(\theta_\mathcal{X})$ The constraints of system state<br>$\mathcal{U}(\theta_\mathcal{U})$ The constraints of applied input |

It is difficult to track the paths effectively by using the model-based controller under uncertain environments. In addition, it is unrealistic to adjust the controller parameters manually for all the conditions. In order for the aforementioned problems to be solved, the data-driven ML methods described in Section 3.1 are usually used to estimate the environmental disturbance, and the data-driven model is used to design the controller. However, the lack of data can easily lead to incorrect approximation of uncertain disturbances. Therefore, the adaptive predictive control strategy is the key to achieving high-performance path tracking control. The recorded data are converted to the corresponding MPC parameters. On one hand, the closed-loop performance is improved by adjusting the parameterized cost function and constraint conditions on the basis of the performance-driven controller learning. On the other hand, the path tracking controller adjusts its parameters online by adapting to uncertain environmental disturbances. The input and state trajectories of the dynamic system are parameterized by the basis function to reduce the computational complexity of MPC [98]. When the input constraints are considered, the uncertainty is estimated and compensated in the design of the controller, and the recorded data are converted to the corresponding parameterization of the MPC problem.

The performance-driven controller learning focuses on finding the parameterization of the cost function and constraint conditions in MPC. The closed-loop performance is optimized by solving the optimal control problem. One solution is to adjust the controller on the basis of Bayesian optimization, and the other solution is to learn the terminal components to counteract the finite-horizon nature of the controller.

The Bayesian optimization method models the unknown function as a GP, evaluates the function, and guides it to the optimal position. Then, the controller parameters are estimated on the basis of experiments to optimize the controller performance. The information samples in Bayesian optimization are usually far away from the original control law, which leads to unstable evaluation and system failure of the controller in the early optimization process [99]. The security problems are solved by using the improved security optimization controllers and exploring new controller parameters whose performance lies above a safe performance threshold [100]. By combining the linear optimal control with Bayesian optimization, a parameterized cost function is introduced and optimized to

compensate the deviation between real dynamics and linear predictive dynamics, resulting in an improved controller with fewer evaluations [101]. The controller is designed using the current linear dynamic model and the parameterized cost function on the basis of the Bayesian optimization. This method evaluates the controller performance gap with the actual physical device in the closed-loop operation, and iteratively updates the dynamic model on the basis of this information to improve the controller performance [102]. The inverse optimization control algorithm is used to learn the appropriate cost function parameters of MPC from the human demonstrated data [103]. The motion generated by the path tracking controller matches the specific characteristics of human-generated motion and avoids massive parameter adjustments. In the consideration of the online estimation and adjustment of control parameters, a framework composed of real-time parameter estimators and feedback control strategies can improve the path tracking performance in mobile platforms [104].

The terminal set is an important design parameter in MPC. The MPC reduces the bad effects caused by the limitation of the prediction range by using terminal cost functions and constraints. The data are collected through the ML methods to improve the terminal components. A large terminal set leads to a large area, and it is quick and feasible in solving the MPC problem in this area. The state convex hull terminated on the trajectory of the origin is proven to be control invariant [105]. On this basis, a method for constructing a terminal set from a given trajectory solves the optimization problem required for parameterized offline calculation terminal controllers [106]. The final terminal controller is a solution for state-dependent optimization problem. For constrained uncertain systems, a robust learning model predictive controller is used for collecting the data from iterative tasks and estimating the current value function, which meets the system constraints accurately [107]. The terminal security set and the terminal cost function for iterative learning are designed to improve the closed-loop tracking performance of the controller [108]. This method estimates the unknown system parameters and generates high-performance state trajectory at the same time. The iterative learning has been further studied, and a task decomposition method for iterative learning model predictive control has been proposed [109]. This method quickly converges to the local optimal minimum time trajectory as compared to simple methods.

It is necessary for the controller to ensure that the mobile platforms accurately track the predefined path under uncertain disturbances. In order for the influence of uncertain environmental disturbances on parameterized objective functions and constraint definitions to be overcome, the components of the controller are learned and adjusted to adapt the environment, and an improved predictive control strategy is established. The combination of the high-level model predictive path following controller (MPFC) and the low-level learning-based feedback linearization controller (LB-FBLC) is used for nonlinear systems under uncertain disturbances [110], as shown in Figure 7. The LB-FBLC uses GP to learn the uncertain environmental disturbances online and accurately track the reference state on the premise of probability stability. The MPFC uses the linearized system model and the virtual linear path dynamics model to optimize the evolution of path reference targets and provides reference states and controls for LB-FBLC. The deep neural networks with a large number of hidden layers significantly improve the learning process of NMPC control law as compared to shallow networks [111]. The integrated design of model learning and model-based control design obtains the advanced prediction model for MPC cost function, the disturbance state space model satisfying robust constraints, and the robust MPC law [112]. Finally, the data-driven controller effectively deals with the constraints and tracks the desired reference output.

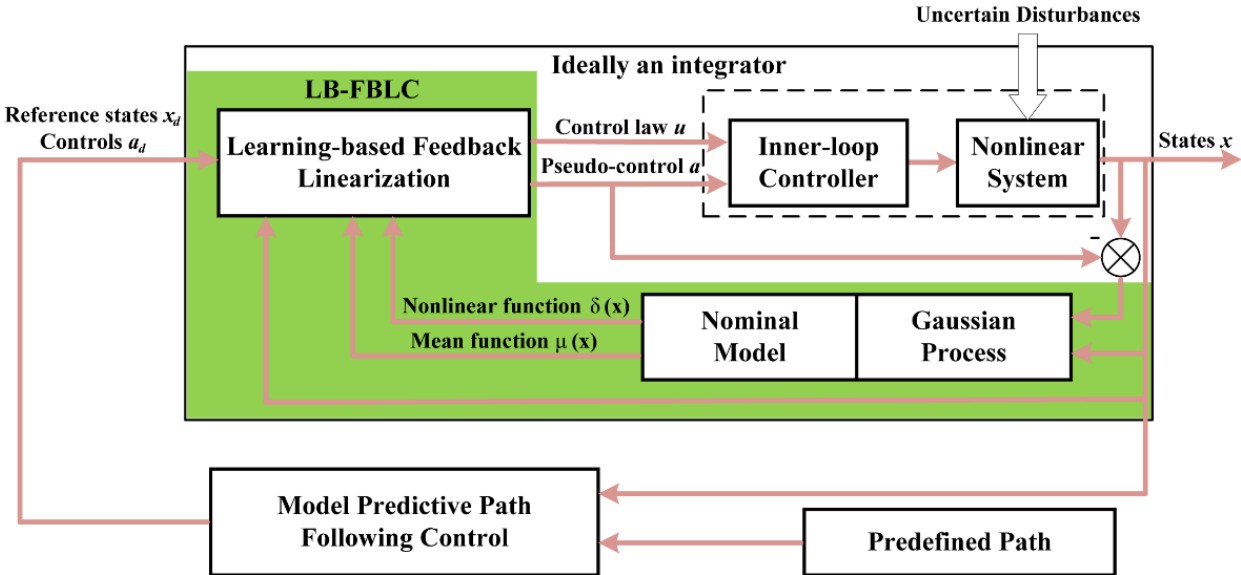

**Figure 7.** The architecture of the proposed strategy for nonlinear system path under uncertain disturbances [110].

### 3.2.2. Learning and Optimization for Controller Real-Time Response

In addition to ensuring the control precision of path tracking control, the controller should also improve the computational complexity. The method for generating a neural network controller with MPC training samples solves the problem of poor real-time response of MPC controllers while ensuring control precision [113]. However, the training samples used in the aforementioned method lose the feedforward information. This problem can be solved by reducing the number of control steps of the controller and using a point-to-line NMPC neural network (PLNMPC-NN) in articulated vehicle path tracking control [114]. The PLNMPC method uses the position and state errors between the predictive horizon and on the reference path as penalty terms and generates the training samples in the non-global coordinate system. The training process and training parameters of PLNMPC-NN are presented in Figure 8.

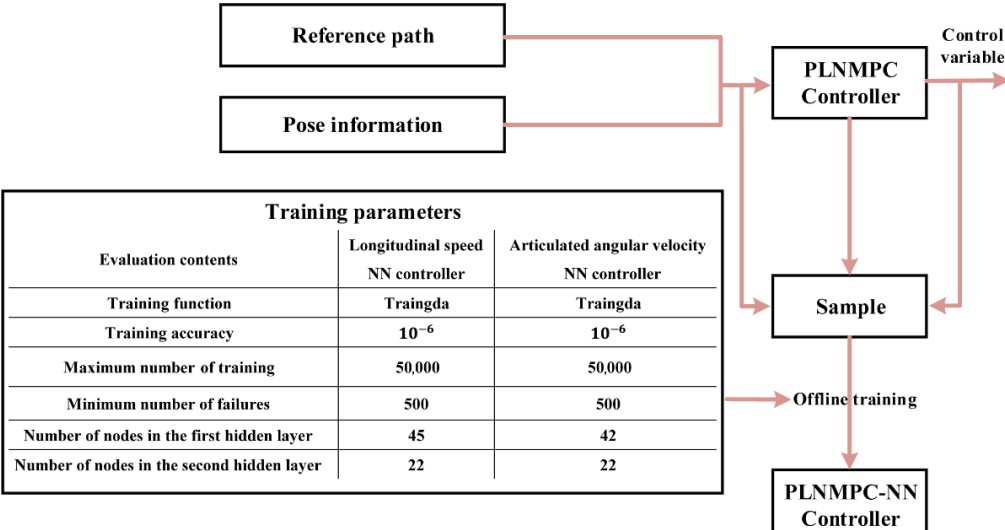

**Figure 8.** The PLNMPC-NN training process and its training parameters [114].

The online update of the recurrent neural network (RNN) models captures the nonlinear dynamics when the model is uncertain. In [115], the RNN method is applied to a

chemical process with time-varying disturbances under LMPC and LEMPC. A real-time control Lyapunov barrier function-based model predictive control (CLBF-MPC) system is developed in the aforementioned research [116]. The CLBF-MPC system considers time-varying disturbances to ensure the closed-loop stability and operation safety, which proves its effectiveness in dealing with the problem of ML model online updating in real-time control.

The reinforcement learning (RL) learns the optimal control command on the basis of the predefined reward function, simulates the real environment in advance, and improves the real-time control process response in the real environment. The RL network is trained offline, and a large volume of offline training data is used for the RL agent, including initial positions, headings, and velocities of the vehicle [97]. The evaluation of the real vehicle shows that the trained agent steadily controls the vehicle and adaptively reduces the speed to adapt the sample path. Such a control process has better real-time performance.

It is noteworthy that the learning and optimization of controller parameters significantly influence the control precision. The offline training of samples is a very effective way to improve the real-time response of the controller. The controller design based on learning methods improves the performance of the path tracking controller itself.

### 3.3. Learning and Optimizing for Controller Output under Uncertain Disturbances

When the uncertain disturbances act on the mobile platforms, the predicted position of the controller may be different from the actual tracking position. Most of the path tracking control methods are unable to ensure that security constraints under physical limitations, especially during learning iterations. It is necessary to choose soft output constraints instead of hard constraints on the input and the rate for achieving more accurate path tracking in mobile platforms.

### 3.3.1. Learning and Optimizing Controller Output Based on Reinforcement Learning

When the mobile platform is driving at a high speed in a known environment or driving in an unknown complex terrain, the MPC controller is unable to slow down the mobile platforms considerably during steering. It is necessary to optimize the output of the MPC controller. The reinforcement learning algorithm evaluates the feedback signal of the environment to improve the action plan and adapt the environment in order to achieve the intended goals. The adaptive MPC path tracking controller based on reinforcement learning is designed to correct the predicted output of the model by interacting with the real environment. The reinforcement learning algorithm is used to adaptively adjust the MPC controller online to realize path tracking in mobile platforms with high robustness on complex terrains.

As compared with the traditional MPC controller, the adaptive MPC path tracking controller based on reinforcement learning further reduces the system overshoot and oscillations. It optimizes the performance of system dynamics and steady-state error. The hierarchical reinforcement learning method improves the generalization ability of reinforcement learning for optimal path tracking in wheeled mobile robots [117]. The behavior–reward scoring mechanism of reinforcement learning is used to learn the behavior rules so that the unmanned surface ships are able to estimate the best path tracking control behavior [118]. The proximal policy optimization (PPO) method is used as a deep reinforcement learning algorithm and combined with the traditional pure pursuit method to construct the vehicle controller architecture [119]. The blend of such controllers makes the overall system operations more robust, adaptive, and effective in terms of path tracking. Similarly, the PPO method is also used to achieve the trade-off between smooth control and path errors by designing the reward function to consider the smoothness and tracking error [120]. Figure 9 presents the flowchart of PPO control framework. The trained model can be nested in the combined controller to improve the accuracy of path tracking control on the basis of the interactions with the environment.

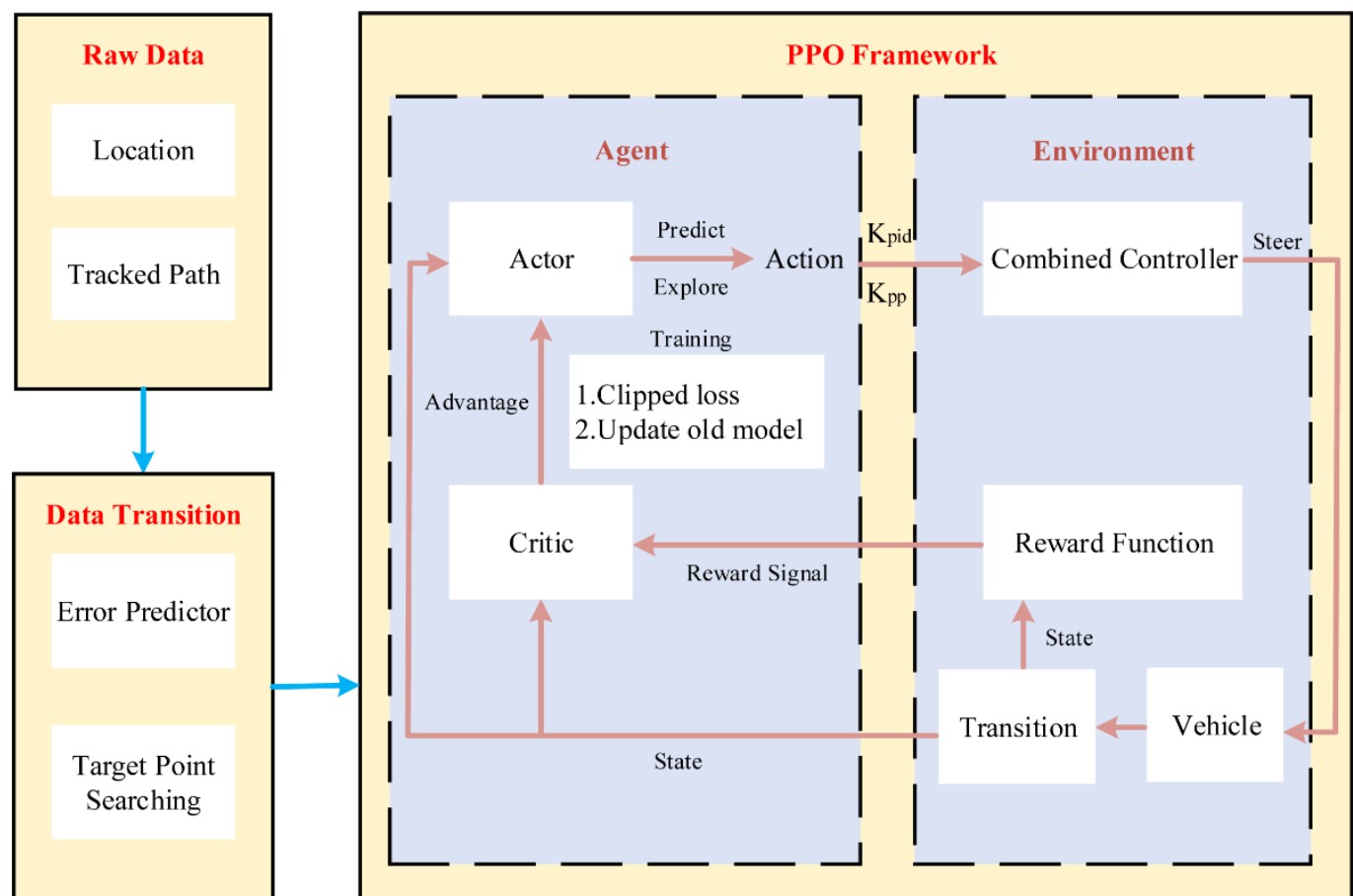

**Figure 9.** The control framework of PPO [120].

On the basis of the aforementioned works, we propose a reinforcement learning-based model predictive control (RLB-MPC) path tracking control framework. Additionally, the output of the controller is optimized on the basis of the reinforcement learning model. First, the predictive path is obtained by controlling the data-driven prediction model in the path-tracking controller. The target path is obtained by using visual and radar sensors. Second, the path deviation generated by the comparison between the predicted path and the target path is considered as the optimization objective. Finally, the path-tracking controller interacts with the real environment on the basis of the reinforcement learning model and corrects the path deviation generated by the comparison between the actual path and the optimized predicted path. The state information that conforms to the internal reward and punishment function of the reinforcement learning model is returned to the reference output and output directly. The other state information is returned to the MPC internal optimization until it meets the requirements of direct output. The RLB-MPC control framework is shown in Figure 10.

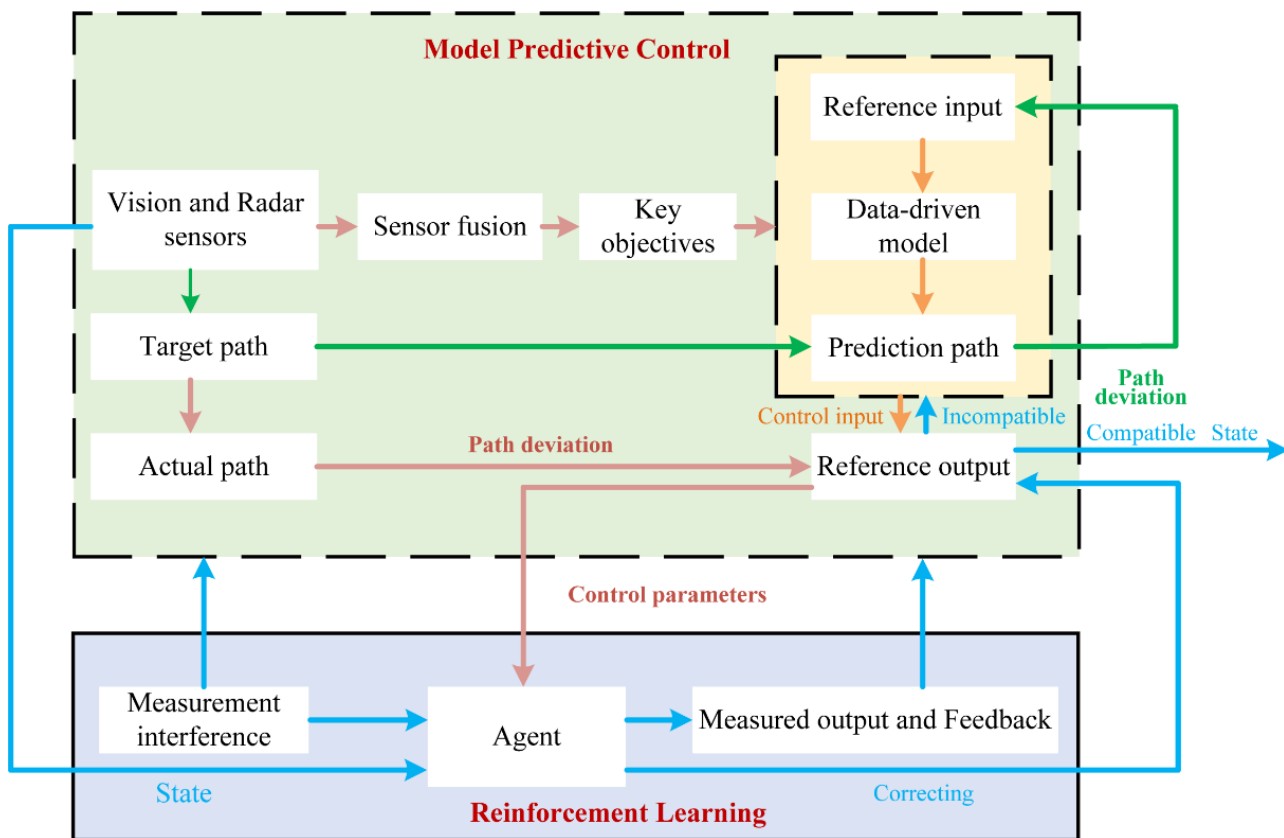

**Figure 10.** The control framework of RLB-MPC.

### 3.3.2. Learning and Optimizing Controller Output Based on Security Framework

The complex and high-dimensional control performance is achieved without the prior knowledge of the system by general learning control technique, especially the reinforcement learning control technique [121]. However, most of the learning control techniques are unable to ensure the security constraints with the physical limitations, especially during the learning iterations. In order for the aforementioned problem to be solved, the security framework was proposed in the control theory [122].

The centralized linear system security framework ensures the security by matching the learning-based input with the initial input of the MPC law online. The MPC law drives the system to a known secure terminal set. The model predictive security filter is applied to control the state and input space [123], as shown in Figure 11. The filter transforms the constrained dynamic system into an unconstrained security system, and any reinforcement learning algorithm can be applied in the security system without any constraint. The security system is established by constantly updating the security strategy. The security strategy is based on MPC formulation using a data-driven prediction model and considering state- and input-dependent uncertainties. The filter ensures the vehicle security during aggressive maneuvers, which was demonstrated by the applications for assisted manual driving and deep imitation learning using a miniature remote-controlled vehicle [124].

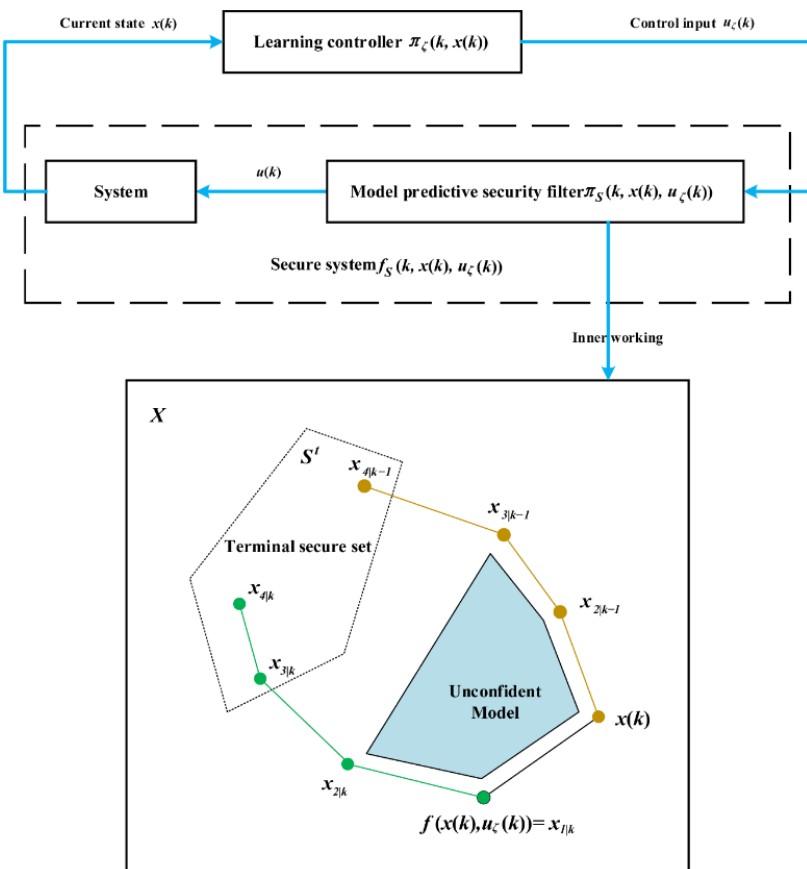

**Figure 11.** The model predictive security filter. On the basis of the current state $x(k)$, a learning-based algorithm provides a control input $u_\zeta(k) = \pi_\zeta(k, x(k))$, which is processed by the security filter, i.e., $u(k) = \pi_S(k, x(k), u_\zeta(k))$, and is applied to the real system. The detailed working of a model predictive security filter is presented at the bottom of the illustrations, which shows the state of the system at time $k$ with a secure backup plan for a shorter horizon obtained from the solution at time $k$-1 (depicted in brown), as well as areas with poor model quality (depicted in green). An arbitrary learning input $u_\zeta(k)$ can pass through the model predictive security filter if a feasible solution towards the terminal secure set $S^t$ is obtained (depicted in green). If this new backup solution cannot be found and the planning problem is infeasible, the system can be driven to the secure set $S^t$ along the previously computed trajectory (depicted in brown) [123].

As compared to the centralized linear system security framework, the distributed model predictive security certification scheme ensures the state and input constraint satisfaction when applying any learning-based control algorithm to an uncertain distributed linear system with dynamic couplings [125]. In addition, two different sets of distributed system security have been proposed on the basis of the latest research findings regarding structural invariant sets [126]. Different sets are different in their dynamic allocation of local sets and provide different trade-offs between the required communication and the realized set size. The synthesis of a security set and control law offer improved scalability by relying on the approximations based on convex optimization problems [127]. For nonlinear and potentially larger-scale systems with security certification, a security framework is proposed to improve the learning-based and insecure control strategies. Furthermore, a probabilistic model predictive security authentication that can be combined with any reinforcement learning algorithm is proposed, relying on the Bayesian scheme. This model provides security assurance in terms of state and input [128].

The control of complex systems faces a trade-off between high performance and security assurance, which in particular limits the application of learning-based methods in security-critical systems. Ensuring the security constraints under physical limitations

and improving the performance of mobile platforms during the learning process are important to the security control system. The LB-MPC can be used in common situations where the model prediction security filter has uncertainties and needs to learn from data. The general security framework based on Hamilton–Jacobi reachability methods uses approximate knowledge of the system dynamics to satisfy the constraints and minimize the disturbances during the learning process [129]. The Bayesian mechanism is further introduced to improve the security analysis by obtaining new data through the system. The reachability analysis is combined with ML [130]. The reachability analysis maintains the security performance, and the ML improves the system performance. When both control inputs and disturbances are bounded, a secure mandatory control action is required only when the system approaches the boundary of the insecure set. At other times, the system can freely use any other controller. The statistical identification tools are used to identify better prediction models that can deal with state and input constraints and optimize the system performance on the basis of the cost function [131]. The statistical model of the system is established by using the regularity assumption of GP on dynamics [132]. More data are collected from the system and the statistical model is updated, improving the control performance and ensuring the security of the learning process.

The proposed security framework removes the security constraints of learning-based control under physical limitations and optimizes the controller by using the model predictive security filters. The combination of model predictive security filters and different types of learning-based control methods not only meets the high-performance requirements of complex systems but also ensures the security in the control process.

## 4. Future Research Challenges

For the interpretability of the prediction model, the prediction model built using the ML method is a kind of black box model and is applicable to specific small-scale environment. However, once the learning fails, it is difficult to ensure the system security based on ML only. Most of the traditional physical models are simplified, linearized, and theoretical. However, the complex modeling process is time-consuming, and the model has calculation errors. The model built by using the LB-MPC belongs to the gray-box model. When the advantages and characteristics of ML and MPC are combined, the LB-MPC method achieves interpretable optimal performance. However, it has poor adaptability to the path without learning tests.

For the accuracy of the prediction model, the traditional prediction model simplified by expressions is not comprehensive because it does not consider the influence of uncertain disturbances. The data-driven prediction model is constructed on the basis of the data generated by the actual operations in mobile platforms. It has good adaptability and higher accuracy in the special working conditions. However, the data-driven prediction model requires a large number of labeled data, and the reinforcement learning model needs to interact with the environment. In order for the unexpected states to be dealt with, both prediction models require a large amount of training time. Reducing the volume of data required by the data-driven prediction model and promoting greater adaptive capacity are challenges in the future.

For controller design, only small random uncertainties can be handled in the mobile platform path tracking controller design, and all behavioral responses under uncertain disturbances cannot be considered. It is necessary to continuously learn from the uncertain disturbances during the operation process in order to improve the performance of the controller.

For the real-time capability of LB-MPC, the data may not capture the mobile platform operation characteristics when it is controlled by MPC. The mobile platform model also requires continuous updating to capture the changes of some physical characteristics over time due to changes in external and internal factors, such as weather, complex curvature variation conditions, mechanical friction, and fatigue failure. However, the automatic model update mechanism is an important challenge for the LB-MPC systems proposed in

previous studies. The way to solve the challenge in the future is to develop a self-adaptive ML-based prediction model that exploits online mobile platform operation data to update the prediction model continuously as the mobile platforms are controlled by MPC in real time.

For the optimization of the controller, the input and output constraints limit the path tracking the performance in mobile platforms. The current security control framework combines the model predictive security filters with different types of learning-based control methods. It solves the trade-off problem between high performance and security assurance faced by the control of complex systems. In order for the performance of path tracking controllers to be further improved under uncertain disturbances, the seeking of more reasonable optimization methods and addressing input and output constraint methods have great research potential.

## 5. Conclusions

The present work reviews the LB-MPC technique and its application in the field of mobile platforms for path tracking control. The LB-MPC and its two components, namely, MPC and ML, are summarized. According to the relevant literature and research results, the application of the LB-MPC in path tracking is classified, and the characters and advantages of the application is explained. Under uncertain environmental disturbances, the data-driven prediction models obtained by LB-MPC can better adapt to complex situations. In controller design, the parameterized version of the LB-MPC problems and offline training samples are introduced to ensure control precision and real-time response. Moreover, combined with the security control framework, the controller output can be optimized in path tracking control. This work also highlights the current research challenges of prediction model interpretability and accuracy, controller design, and output optimization in LB-MPC. It will provide a reference for the research and application of LB-MPC in path tracking control in mobile platforms.

**Author Contributions:** Conceptualization, K.Z., J.W. and X.X.; data curation, X.L.; funding acquisition, J.W; methodology, K.Z., J.W. and J.H.; software, C.S. and W.K.; supervision, X.L.; project administration, X.X. and X.L.; writing—original draft preparation, K.Z., J.W. and X.X.; writing—review and editing, K.Z., J.W. and X.L. All authors have read and agreed to the published version of the manuscript.

**Funding:** This research was funded by the National Natural Science Foundation of China, grant number 51875239.

**Institutional Review Board Statement:** Not applicable.

**Informed Consent Statement:** Not applicable.

**Data Availability Statement:** The data used to support the findings of this study are available from the corresponding author upon request.

**Acknowledgments:** The authors would like to thank all anonymous reviewers and editors for their helpful suggestions for the improvement of this paper.

**Conflicts of Interest:** The authors declare no conflict of interest.

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
