# Peer review of "A Survey on Learning-Based Model Predictive Control: Toward Path Tracking Control of Mobile Platforms"

_applsci, doi:10.3390/app12041995_

Round 1

Reviewer 1 Report

The idea looks good. However, several remarks should be considered: 

The motivation should be explained more clearly. 

The whole paper should be re-organized; I propose to divide the paper into major paragraphs: Introduction, Problem formulation, prior work, Proposed method, Experimental results and discussion, and conclusion.

All references must be revised. Issue numbers and volume numbers are missed from most of the references.

The authors need to highlight their novelty contribution. 

The introduction section should be extended.
In the introduction section, the authors should focus on the main issues here. Background, the problem statement, the motivations behind the work and its context, the main contributions, and the outlines of the paper.

The outlines of the paper must be written as the last paragraph of the introduction section.

The related work should be written as section 2 after the introduction with more recent references to highlight the contribution of this paper.
The proposed method is inadequately described. Better start by providing the reader with a high-level picture of the problem.
There is no analysis of the extracted results and no discussion.

Reviewer 2 Report

The manuscript is a comprehensive survey of learning-based MPC methods applied to the problem of path-tracking of mobile platform. The paper introduces the problem, the machine learning methods and describes a large variety of methods. The paper is written in a clear fashion, however the language could be improved.

I have only a couple of general comments:

  • A discussion of the different mobile platforms where the described models are applied is missing.
  • The descriptions of the methods are often high-level and I believe it requires some more mathematical detail with concrete equations.
  • The authors need to give more details on the numerical examples used to obtain the results in Figures 2 and 3.

Reviewer 3 Report

This paper proposes a so-called learning-based model predictive control method for trajectory following tasks. The work may have some merits but generally, this manuscript is not at all organized and is trying to address too many challenging problems at once, without presenting any concrete solution to even one of them. 

  1. To deal with MPC, one needs to show what model it is and how the params can be predicted. Obviously, this critical point is missing from the work. The authors claim they propose a learning-based method, while the reviewer finds no hint about how the dataset and label are made.
  2. The proposed method can be summarized, as the authors claim, using Fig.3. However, the reviewer is not clear about how the cycle divisions can be modelled using LSTM and how the regression can be performed after LSTM. No further information is provided other than this picture. Therefore the reviewer doubts the authenticity of the work conducted.
  3. As an academic paper other than a survey paper, this work does not show any validation or experiments to justify the performance of the proposed method, which is quite odd. 

The reviewer strongly recommends the authors rewrite the manuscript and show the contribution clearly before the next submission.

Reviewer 4 Report

The paper presents a summary of the literature on Learning-Based Model Predictive Control, with a claimed focus on path tracking of mobile robots.

The paper is not easy to follow in my opinion, thus the actual contribution is unclear. While it is clear that the paper is in the end a literature review, such a literature is not classified according to a systematic approach, which is expected to be more adequate when a very large number of references are included and a specific application domain is meant to be addressed, as it is for the paper under review. I would suggest to be more explicit from the beginning in describing the organization of the literature into classes of proposed methods and related applications and the re-structure the following sections accordingly.

Anothe aspect that is not clear is whether or not the authors are proposing a new or recently proposed method of theirs, since the paragraph at the end of page seems to be written with this purpose, but the novel contribution of this specific paper never appears. The same paragraph ends mentioning "two different materials" that are not clearly explained in the rest of the paper.

Finally, while I have appreciated the intense use of block diagrams to present most of the different reviewed methods, I am not convinced that the content of Table 2 is really useful, since the equations there are not so easy to understand without a knowledge of the related references and especially the notations are not fully explained. The table should be revised to explain all the mathematical symbols or deleted at all (which is probably preferable)

Round 2

Reviewer 1 Report

The authors addressed all previous comments.

I recommend the paper to be accepted in this round.

Author Response

Thank you very much for your efforts and advice.

Reviewer 3 Report

The reviewer appreciates the effort the authors made in revisions. Since this manuscript is a review paper, the reviewer suggests two more revisions to this work:

  1. Revise the title to be "A Survey on..." instead of the current one.
  2. The review content needs significant improvements, including the depth of review, the significance of the reviewed research domain (Learning-Based Model Predictive Control), the remaining challenges of the reviewed research domain. The current work only focuses on too high-level review, i.e., machine learning and MPC, along with a few applications using Learning-Based Model Predictive Control, without technical depth. Thus the reviewer believes the current work is not sufficient to make a survey paper.

Reviewer 4 Report

The authors have addressed the issues raised by the reviewer. 

Author Response

(The authors gave the same response as above.)
